# Response Time Improves Gaussian Process Models for Perception and Preferences

**Michael Shvartsman**[1]          **Benjamin Letham**[1]          **Eytan Bakshy**[1]          **Stephen Keeley**[2]

[1]Meta,
[2]Fordham University, Department of Natural Sciences, New York, NY

## Abstract

Models for human choice prediction in preference learning and perception science often use binary response data, requiring many samples to accurately learn latent utilities or perceptual intensities. The response time (RT) to make each choice captures additional information about the decision process, but existing models incorporating RTs for choice prediction do so in a fully parametric way or over discrete inputs. At the same time, state-of-the-art Gaussian process (GP) models of perception and preferences operate on choices only, ignoring RTs. We propose two approaches for incorporating RTs into GP preference and perception models. The first is based on stacking GP models, and the second uses a novel differentiable approximation to the likelihood of the diffusion decision model (DDM), the de-facto standard model for choice RTs. Our RT-choice GPs enable better latent value estimation and held-out choice prediction relative to baselines, which we demonstrate on three real-world multivariate datasets covering both human psychophysics and preference learning.

## 1 INTRODUCTION

Human binary choice data are widely used to measure latent mental constructs. Key motivating applications are human psychophysics, the study of human perception [Kingdom and Prins, 2016]; human value-based decision making [Rangel et al., 2008]; and preference learning [Fürnkranz and Hüllermeier, 2003]. In all cases, humans give binary responses about whether they detect a stimulus or can discriminate between two stimuli (in psychophysics), or about which of two options they prefer (in value-based decision-making and preference learning). Although binary choice experiments have been used in psychology for more than a century [e.g. Fechner, 1860], they have seen recent advances in the machine learning community, particularly through nonparametric latent function modeling and active learning. Since Chu and Ghahramani [2005], Gaussian processes (GPs) have been a standard approach in preference learning for modeling latent utility functions from binary preferences expressed over general multivariate and continuous feature spaces. Among their many applications, human preference data has been used to learn robot locomotion policies [Tucker et al., 2021, 2022, Cosner et al., 2022], personalize assistive devices [Thatte et al., 2017, Tucker et al., 2020], and learn a good golf swing [Biyik et al., 2020]. Recent work in machine learning for psychophysics has similarly used GP models to learn latent perceptual functions from binary human feedback, for purposes including audiometry [Gardner et al., 2015a,b], measuring visual sensitivity [Letham et al., 2022], and understanding perception in augmented/virtual reality devices [Guan et al., 2022, 2023].

In these applications, the model assumes that binary responses derive from a latent function on the input space that is mapped to choice probability through a sigmoidal link function. There are two important aspects of the problem that are detrimental to the sample efficiency of the model. First, information is lost for large portions of the latent space that are mapped to choice probabilities very near 1 or 0. Second, areas of the function with high uncertainty, that is, where preference or detection probability is close to 0.5, require many samples for accurate estimation since they have a large Bernoulli variance, $p(1-p)$. These shortcomings are due to the fact that binary responses are a very coarse measurement of the underlying continuous function reflecting a human's decision process. A richer model of the decision process should allow for discrimination between a 'yes' response with choice probability close to 0.5 and one close to 1.

Psychology and neuroscience provide rich models for the underlying decision process. These models incorporate additional information, notably response times (RTs), as a way of inferring the latent function underlying the subject's re-

*Accepted for the 40th Conference on Uncertainty in Artificial Intelligence* (UAI 2024).

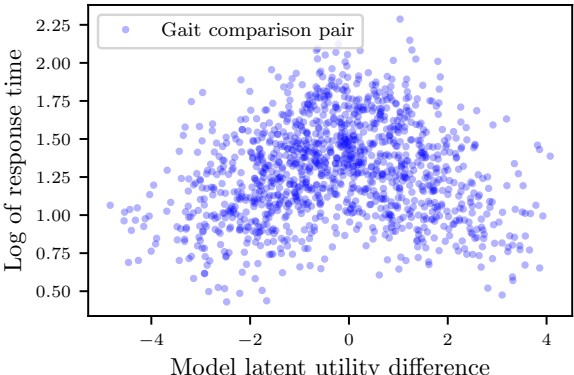

Figure 1: A human subject evaluated 1,225 pairs of robot gaits to select the more natural looking gait. The response time in making the judgement was longer for pairs that have a small difference in their latent utilities, reflecting the increased challenge of judging gaits of similar quality. Response time is useful auxiliary information for learning the latent utility function and predicting choices.

sponse [e.g. Laming, 1968, Donders, 1969, Sternberg, 1969, Clithero, 2018]. One of the most popular such models is the diffusion decision model, also called the drift-diffusion model (DDM) [Ratcliff, 1978, Bogacz et al., 2006, Ratcliff and McKoon, 2008]. Unfortunately, the joint choice-RT likelihood of the DDM cannot be computed in closed form. A variety of numerical approaches can be used to approximate it [Voss and Voss, 2008, Navarro and Fuss, 2009], but none are differentiable. This prevents them from being incorporated into a modern variational GP approximation framework in a straightforward way. Our core contribution is to approximate the DDM using a family of parametric skewed distributions, which enables for the first time the use of GP models with DDM-inspired RT-choice likelihoods.

To illustrate the relationship between RTs and choice, Fig. 1 shows RT data from the multivariate robot gait optimization task we study in Section 7.1, in which a human subject watched two simulations of a quadruped robot walking, each with different gait parameters, and was asked which gait looked more natural. The figure compares the latent GP utility estimates for each evaluated pair using the binary preference data only ('choice-only' model), with the response time of the human subject in judging that pair. When the difference in latent utility for the pair is 0, they are equally preferred, and the choice probability is 0.5. Gaits with closer utility values had both longer and more variable response times, while those with large differences in utility were easier to judge and had shorter and less variable response times. This is the relationship that we use to improve latent function estimation and choice prediction.

RTs are particularly valuable as an implicit measure of confidence because they are easily recorded alongside binary

choice decisions in experiments with humans. They allow us to improve model performance without having to change from experimental designs already in use, and without burdening the subject with additional questions to elicit explicit confidence assessments. This is especially valuable in the human user studies that are the focus of this work, where minimizing load on the subject is of paramount importance.

Alongside the psychology-driven approach using the DDM likelihood, we also propose a simple stacking approach for incorporating RTs into a choice GP, in which a GP model for RTs is used as an input to a second layer GP that models choice. We show that the DDM-augmented model consistently outperforms the choice-only baseline, but does so far better in the regime where DDMs are conventionally used (accurately-measured, non-deliberative short decisions). When RT measurements are lower quality or consistency, the more flexible stacking model can perform better.

We study the performance of the models using both synthetic data and data from real human subject studies. On synthetic problems, we show that leveraging RTs provides more accurate estimation of the latent function than choice-only models, especially in realistic low-data regimes. On real data, we show that incorporating RTs into GP models can substantially improve choice prediction performance relative to choice-only models. This is the case even when the choice probability is the only quantity of direct interest and the RTs are solely used as side information for the modeling, as in machine learning applications in this domain.

Section 2 provides background on RT modeling and the DDM. Section 3 introduces the GP classification model used for modeling human choices. Section 4 then describes our novel DDM approximation and how we use that to jointly model RTs and choices in a GP, as well as the stacking model. Section 5 describes the synthetic experiments, followed by the real-world psychophysics and preference learning experiments in Sections 6 and 7 respectively. Our psychophysical dataset is from a high-dimensional visual psychophysics task. Our first preference learning example is a novel robotics preference learning dataset[1], the robot gait optimization used for Fig. 1. Our second preference learning dataset comes from a study of recommender system evaluation, containing pairwise evaluations of A/B test outcomes at an internet company.

---

[1]This new dataset, alongside code used to generate the figures in the paper, is at https://github.com/facebookresearch/response-time-gps. The core modeling and estimation code will be available as part of the AEPsych package for adaptive experimentation for human experiments (https://aepsych.org/).

## 2 BACKGROUND

The DDM is widely used for modeling decision making in neuroscience and psychology, and can be motivated from a variety of theoretical perspectives: as a generalization of classical signal detection theory in psychophysics [Ashby, 1983, Griffith et al., 2021], as a sequential statistical inference process [Bogacz et al., 2006], as an approximation to neural firing rates [Gold and Shadlen, 2002, 2007], or as a mechanistic theory of memory [Ratcliff, 1978]. With just a few parameters, the model describes the joint distribution of choices and RTs. The RT is generally understood to reflect a process of evidence accumulation, sequential statistical inference, or integration over neural noise. When this process reaches some threshold determined by the desired accuracy of the decision maker, a decision is made. The process completes more quickly when stronger signal is available, resulting in faster decisions when signal is stronger.

Existing DDM models almost universally estimate parameters independently over a set of discrete experimental conditions, making them incompatible with the general continuous stimulus spaces that are of interest here. The RT distribution under the DDM is that of the first-passage time of a 1-d Wiener process with nonzero drift and nonzero initial condition to one of two boundaries. While expressions for this distribution are well-known [Feller, 1966], they take the form of an infinite summation. This sum can be truncated while controlling approximation error [Navarro and Fuss, 2009], but naive application of this approximation is incompatible with modern differentiable programming frameworks, for two reasons: first, because bounding density error does not necessarily bound error in gradients; and second, because varying term counts per parameter value preclude leveraging standard batched linear algebra operations. Alternate approaches solve the Kolmogorov backward equation associated with the DDM process [Voss and Voss, 2008, Voss et al., 2015, Shinn et al., 2020] or approximate parameters by moment-matching to the data [van Ravenzwaaij et al., 2017]. Given this complexity, standard approaches to DDM estimation rely on full MCMC using slice sampling [e.g. Frank et al., 2015] or zeroth-order optimization.

Our focus is not necessarily improving DDM likelihood approximation or density estimation. Rather, we would prefer to use a simpler density that is still able to represent the latent value or signal strength we need for choice prediction in a GP framework or other ML models. Unfortunately, while other distributions have been used to describe response times, their parameters do not map to the domain knowledge encoded in the DDM process in a straightforward way [Matzke and Wagenmakers, 2009]. Instead of using such distributions directly, we use the fact that closed-form expressions for the conditional moments of the DDM distribution are known even if the exact density is intractable [Srivastava et al., 2016]. We use these moments, which are a function

of the DDM parameters, to match the moments of a shifted, skewed distribution with a known functional form such as the shifted lognormal or shifted inverse gamma distributions. We select the parameters of these three-parameter distributions to uniquely match the mean, variance, and skew of the DDM distribution [Lo et al., 2014].

## 3 GP MODELS FOR HUMAN CHOICES

GP models can successfully model both human perception [Gardner et al., 2015a, Owen et al., 2021, Letham et al., 2022, Keeley et al., 2023] and preferences [Chu and Ghahramani, 2005, Lin et al., 2022]. Observations are given as $\{\mathbf{x}_n, y_n\}_{n=1}^N$, where $\mathbf{x}_n \in \mathbb{R}^d$ are multi-dimensional stimulus configurations, and $y_n \in \{0, 1\}$ are subject responses. The typical GP approach to modeling in this setting is to assume a latent function $z$ with a GP prior:

$$z(\mathbf{x}) \sim \mathcal{GP}\left(0, k_\theta\left(\cdot, \cdot\right)\right). \tag{1}$$

For single choices (e.g. 'yes' vs. 'no'), the kernel $k_\theta(\mathbf{x}, \mathbf{x}')$ can be a standard GP kernel such as the radial basis function (RBF), which we use throughout our experiments. For preference choices between paired inputs (e.g. 'prefer 1' or 'prefer 2'), $z(\mathbf{x})$ models a utility function, and we assume that the choice probability is determined by the difference in utility between the two choices [Chu and Ghahramani, 2005]. We do this by using the 'preference kernel' given by Houlsby et al. [2011], which exploits the fact that GPs are closed under addition to convert a GP prior over the latent function to a GP prior over the paired differences with a particular kernel formulation. In both cases we estimate hyperparameters controlling the amplitude and an independent lengthscale per input dimension (i.e., an ARD kernel), which hyperparameters we denote $\theta_G = \{\rho, \boldsymbol{\ell}\}$.

The observation model is Bernoulli, and assumes that $y$ is conditionally independent of $\mathbf{x}$, given $z$. Formally, let $z_n = z(\mathbf{x}_n)$, and $y_n \sim \text{Bernoulli}(\Phi(z_n))$, where $\Phi(\cdot)$ is a sigmoid, typically the Gaussian cumulative distribution function. Prior work has varied the choice of the sigmoid and the details of the kernel, but has maintained this basic model structure. We are primarily interested in inferring $z$, both for the purpose of predicting $y$ and for extracting useful information such as detection thresholds and most-preferred inputs. We refer to this model as the 'choice-only' model since it uses only choice data $y_n$. We now show how this model can be extended to incorporate RT observations.

## 4 THE RT-CHOICE MODEL

We augment the GP model above to include a distribution over RTs. Now, our data are $\mathcal{D} = \{\mathbf{x}_n, y_n, t_n\}_{n=1}^N$ where $\mathbf{x}_n$ and $y_n$ are as before, and $t_n \in (0, \infty)$ are the RTs. As before, we assume the corresponding latent function values $z_n$ depend on $\mathbf{x_n}$, and put a GP prior on $z(\mathbf{x})$ as in (1).

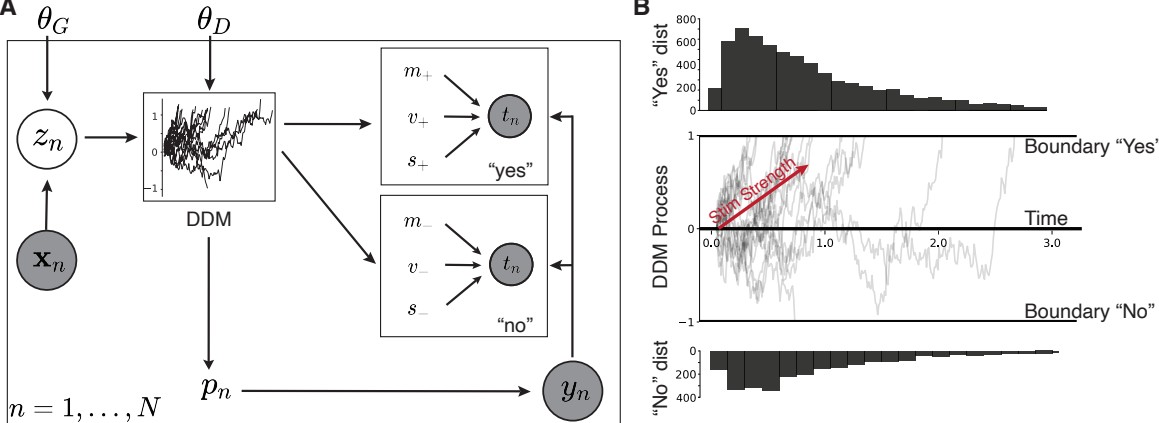

Figure 2: **A.** A graphical depiction of our RT-choice model. The latent variable $z_n$ has a GP prior and is a function of $\mathbf{x}_n$ in the input space. It is used as the drift parameter in the DDM. Via the DDM, it produces RT distributions for both 'yes' and 'no' choices as well as a choice probability $p$. **B.** Schematic of the diffusion decision model. A stochastic process with an average drift (red arrow) dictates random movement in a latent space, capturing an underlying decision making process. The latent accumulator eventually reaches one of two boundaries, representing one of two possible decisions, providing both a choice and a response time. RT distributions for each choice (top and bottom) are skewed with known moments.

Let $\mathbf{y} = (y_1, \ldots, y_N)$, $\mathbf{t} = (t_1, \ldots, t_N)$, $\mathbf{z} = (z_1, \ldots, z_N)$, and $\mathbf{X} = (\mathbf{x}_1, \ldots, \mathbf{x}_N)$. The joint likelihood of RTs and choice responses can be written as:

$$p(\mathbf{t}, \mathbf{y} \mid \mathbf{X}, \theta_G, \theta_D)$$
$$= \int p(\mathbf{t} \mid \mathbf{z}, \mathbf{y}, \theta_D) p(\mathbf{y} \mid \mathbf{z}, \theta_D) p(\mathbf{z} \mid \mathbf{X}, \theta_G) d\mathbf{z}. \quad (2)$$

Here both RTs $\mathbf{t}$ and choices $\mathbf{y}$ are assumed to depend on the input $\mathbf{X}$ only via the latent function $\mathbf{z}$; see Fig. 2A for a graphical representation of the model. The distribution of the latent function values, $p(\mathbf{z}|\mathbf{X}, \theta_G)$, will be Gaussian due to the GP prior on $z$. The choice distribution, $p(\mathbf{y} \mid \mathbf{z}, \theta_D)$, and the conditional RT distribution, $p(\mathbf{t} \mid \mathbf{z}, \mathbf{y}, \theta_D)$, are specified according to a DDM, with parameters $\theta_D$. We will now describe the DDM and these distributions in detail.

### 4.1 THE DIFFUSION DECISION MODEL

The DDM can be simulated as a Wiener process that stochastically moves towards one of two boundaries, the 'yes' boundary or the 'no' boundary. Whichever boundary is reached first is the choice made, $y_n$, and the time required to reach the boundary is the RT, $t_n$. An illustration of the DDM process is shown in Fig. 2B. The movement towards a boundary models the accumulation of evidence, and when the boundary is reached, there is sufficient evidence to make a judgement. The RT is thus the first-passage time of this process, a well-studied quantity in stochastic processes.

The DDM contains several parameters: drift rate, the decision threshold level ($C$), the initial condition ($x_0$), and a shift ($t_0$). We use the GP latent function value $z_n$ as the drift rate, providing an explicit link between the input $\mathbf{x}_n$ and the

response produced by the DDM. The remaining parameters, $\theta_D = \{C, x_0, t_0\}$, will be directly estimated from data.

The DDM process induces different RT distributions for the 'yes' and the 'no' choices, depending particularly on values of the initial condition and drift parameters, as they favor one choice over the other. The evaluation of the likelihood under these distributions is intractable, but their moments can be analytically calculated as a function of $\theta_D$ and $z_n$ [Srivastava et al., 2016]. The first three moments of the RT distributions are denoted in Fig. 2 as $(m_+, v_+, s_+)$ and $(m_-, v_-, s_-)$ for the mean, variance, and skew of the 'yes' and 'no' distributions, respectively. The DDM RT moments and choice probabilities are then incorporated into the full model likelihood in (2) using the exact DDM choice probability, $p(\mathbf{y}|\mathbf{z}, \theta_D) = \prod_{n=1}^{N} p(y_n|z_n, \theta_D)$ and a moment-matching approach for the RT distributions, $p(\mathbf{t} \mid \mathbf{z}, \mathbf{y}, \theta_D)$. We elaborate on each below.

### 4.2 THE CHOICE DISTRIBUTION

The choice distribution in (2), $p(\mathbf{y}|\mathbf{z}, \theta_D)$, is the probability of yes / no choices given the DDM parameters and latent function value. Unlike the response time, it can be computed exactly. We assume conditional independence across trials, $p(\mathbf{y}|\mathbf{z}, \theta_D) = \prod_{n=1}^{N} p(y_n|z_n, \theta_D)$, and have $y_n|z_n, \theta_D \sim$ Bernoulli($p_n$). The DDM process induces the following link function between the latent function values and the choice probability [Srivastava et al., 2016]:

$$p_n = \frac{e^{2Cz_n} - e^{-2x_0 z_n}}{e^{2Cz_n} - e^{-2Cz_n}}. \quad (3)$$

Fig. 3 shows how this link function compares to the probit and logistic sigmoid link functions that have been used in

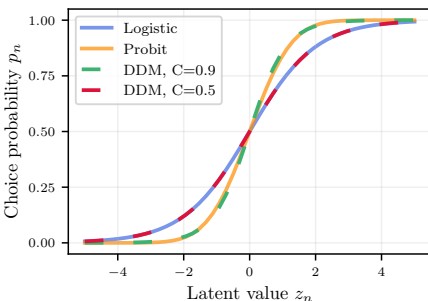

Figure 3: The DDM link function closely matches typical probit and logistic link functions, depending on the process parameters. The supplementary material includes examples of link functions fit to real data.

choice-only GP models, for $x_0 = 0$. It can closely match either depending on the DDM boundary parameter $C$. Therefore, instead of invoking standard link functions (e.g. probit and logistic) in this work, we use the flexible link representation derived from DDM theory.

### 4.3 MOMENT MATCHING THE RT DISTRIBUTION

The conditional RT distributions as a function of choice, $p(\mathbf{t} \mid \mathbf{z}, \mathbf{y}, \theta_D)$, are not available in closed form under the DDM process, however, as discussed above, the moments are. To obtain a tractable likelihood in (2) we will assume conditional independence across trials, $p(\mathbf{t} \mid \mathbf{z}, \mathbf{y}, \theta_D) = \prod_{n=1}^{N} p(t_n|z_n, y_n, \theta_D)$, and will then use a parametric distribution for $p(t_n|z_n, y_n, \theta_D)$, whose parameters are set by moment matching to the DDM RT distribution.

**Proposition 1** (Srivastava et al. 2016). *Let $k_z = Cz_n$ and $\tilde{y}_n = k_z + x_0 z_n (-1)^{(1-y_n)}$. The RT distribution under the DDM process has as its moments:*

$$\mathbb{E}[t_n|z_n, y_n, \theta_D]$$
$$= t_0 + \frac{1}{z_n^2}\Big(2k_z \coth(2k_z) - \tilde{y}_n \coth(3k_z - \tilde{y}_n)\Big),$$

$$Var[t_n|z_n, y_n, \theta_D] = \frac{1}{z_n^4}\Big(4k_z^2 \operatorname{csch}^2(2k_z)$$
$$+ 2k_z \coth(2k_z) - \tilde{y}_n^2 \operatorname{csch}^2(\tilde{y}_n) - \tilde{y}_n \coth(\tilde{y}_n)\Big),$$

$$Skew[t_n|z_n, y_n, \theta_D] = \frac{1}{z_n^6}\Big(12k_z^2 \operatorname{csch}^2(2k_z)$$
$$+ 16k_z^3 \coth(2k_z) \operatorname{csch}^2(2k_z) + 6k_z \coth(2k_z)$$
$$- 3\tilde{y}_n^2 \operatorname{csch}^2(\tilde{y}_n) - 2\tilde{y}_n^3 \coth(\tilde{y}_n) \operatorname{csch}^2(\tilde{y}_n)$$
$$- 3\tilde{y}_n \coth(\tilde{y}_n)\Big).$$

We use these three moments from the DDM to match parametric distributions to the DDM RT distribution. Considering that RT distributions are typically heavy-tailed [Murata

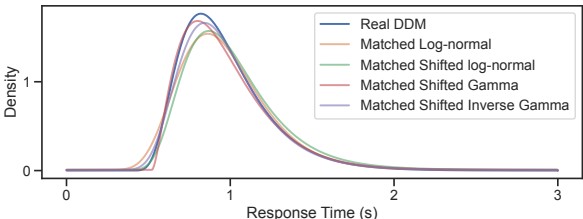

Figure 4: Example of the real DDM distribution (computed using the approximation of Navarro and Fuss 2009), and our moment-matched approximations.

et al., 2014], we focus here on heavily skewed distributions for our parametric RT forms. In our experiments, we use the lognormal, shifted lognormal, shifted inverse gamma, and shifted gamma distributions. In the economics community, expressions are available for the parameters of these distributions as a function of the empirical sample statistics—specifically the mean, variance, and skew [Lo et al., 2014, Brignone et al., 2021]. These expressions allow for analytic moment matching with the known expressions of the DDM RT moments above. To evaluate the likelihood, we use the fitted DDM parameters and GP function samples to compute the mean, variance, and skew of the RT distribution according to Prop. 1. The parameters of the desired parametric RT distribution are then computed from those moments via moment matching, and we evaluate the likelihood of the RTs under this parametric distribution. The formulae for computing the parameters from the moments for each skewed distribution are provided in the supplementary materials. This moment-matched parametric distribution is then used as the RT component of the likelihood in (2), $p(t_n|z_n, y_n, \theta_D)$.

Figure 4 shows how the numerically calculated DDM RT distribution is captured by each of our parameterized heavy-tailed distributions via moment-matching Prop. 1 with the parameter expressions given in the supplement. It is visually apparent that the approximations do not perfectly match the gold-standard series truncation approach (which we use as the 'real' DDM distribution). However, they are overall similar, and we will see below that these approximations are sufficient to enable RT-choice to outperform the choice-only model. We additionally consider 'lapse' RTs, which are thought to be stimulus-independent and arise due to distraction, fatigue, etc. We model such RTs as drawn uniformly from the empirical range of the observed RTs, and parameterize the overall RT distribution as a mixture between the DDM-derived likelihood and this lapse distribution [Ratcliff and Tuerlinckx, 2002].

### 4.4 INFERENCE

Because the marginal likelihood in (2) cannot be computed in closed form, we use standard variational methods to ap-

proximate the GP posterior, and obtain point estimates of $\{\theta^G, \theta^D\}$. Importantly, the parametric moment-matched distributions are all differentiable, so we can compute gradients of the GP hyperparameters and variational approximation with respect to the RT likelihood, rendering the full scheme compatible with modern GP inference tooling. Consistent with standard approaches, we use Gauss-Hermite quadrature in the expectation term of the traditional evidence lower bound, and optimize the objective with gradient-based optimization [Hensman et al., 2015, Balandat et al., 2020]. Estimation takes on the order of seconds on a standard laptop.

## 4.5 A STACKING APPROACH

In addition to the DDM likelihood, we also introduce a simple stacking approach for including RTs into a choice GP. In this approach, we fit two GP models. Let $\tilde{t}_n = \log(t_n)$ be the log RT; the log transform is helpful for enabling GP modeling of the highly skewed RTs. The first model $g$ is a GP regression model fit to the log RT data, modeling $\tilde{t}_n = g(\mathbf{x}_n)$. The second model $h$ is a GP choice model whose input space is augmented with log RT as an additional feature, so that $y_n \sim \text{Bernoulli}(\Phi(h(\mathbf{x}_n, \tilde{t}_n)))$. The kernel over the original input space $\mathbf{x}$ is combined with an RBF kernel over $\tilde{t}$ via a product kernel. These models can be fit independently, but are used together for predicting an input $\mathbf{x}$ for which RTs have not been observed. At prediction time, we take

$$y \sim \text{Bernoulli}(\Phi(h(\mathbf{x}, \bar{g}(\mathbf{x})))),$$

where $\bar{g}(\cdot)$ is a plug-in estimate using the posterior mean of $g$. This model is simple to implement and understand, though there are many real-life properties of RT distributions and their relationship to choices that it does not capture.

# 5 SYNTHETIC EXPERIMENT

To demonstrate the benefits of our approach, we begin with a synthetic data experiment. We use the 2-d detection test function of Owen et al. [2021], which was designed to evaluate models for psychophysics, with the output scaled by a factor of 0.2 to be in the range of typical drift rates in the literature [Matzke and Wagenmakers, 2009]. Fig 5A shows the basic properties of the test function. At every point in the parameter space, the test function (bottom left) was used as the drift parameter of a DDM. From this latent function and the DDM parameters, we can calculate the mean and standard deviation of the RTs for each stimulus using Prop. 1 (top row). The latent function values generate choice probabilities via (3) (bottom right). The bottom right panel also shows the locations of 10 observations, at each of which the choice $y_n$ and RT $t_n$ were obtained by full simulation of the DDM process (Sec. 4.1). The simulated choice $y_n$ is shown

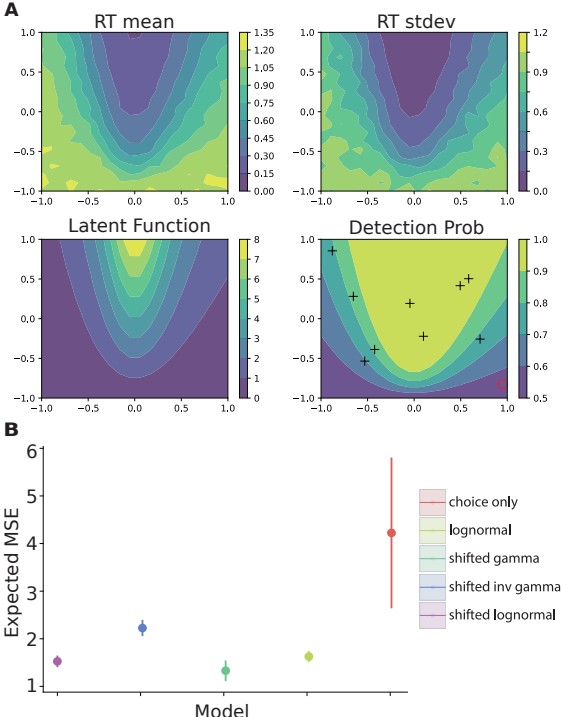

Figure 5: **A.** The mean and standard deviation (top) of RT in our 2-d test function, the latent function value (bottom left), and associated choice probabilities (bottom right). **B.** Mean squared error in expectation over the posterior of the latent function under each model fit on 10 observations, for the choice only model and four DDM-based RT-choice models, using different parametric forms of the RT distribution. Error bars show standard error over 10 simulated datasets.

for these observations in the figure, indicated with a '$+$' for a correct detection and '$\circ$' for detection failure.

Note that there is only a single negative response (detection failure) in this example, a common occurrence in the low-data regime in such problems. In this case, a choice-only model cannot do much more than separate the space into broad 'yes' and 'no' regions, whereas a model taking advantage of RTs can do much more. Fig. 5B shows error on recovering the true, latent function from only 10 observations, in expectation over the GP posterior. All variants of the RT-choice model, using different moment-matching distributions, far outperform the choice-only model. Moreover, the choice-only model's performance is highly variable as it strongly depends on the presence of sufficiently balanced numbers of 'yes' and 'no' trials. We will see this same significant advantage for RT-choice models in the low-data regime in the real-world problems as well.

# 6 REAL-WORLD PSYCHOPHYSICS

As a first evaluation of our model in a real-world setting, we fit the model to data from a high-dimensional visual psychophysical task. These data consist of 1,500 trials from a two-alternative forced choice (2AFC) task provided by Letham et al. [2022], and we obtained RTs from the authors. For each trial, the subject was shown an animated Gabor patch, one half of which had been scrambled, with the scrambled side selected randomly each trial. The subject was asked to identify which side was not scrambled. The stimulus in each trial varied along six dimensions (contrast, background luminance, temporal and spatial frequency, size, and eccentricity), rendering some trials harder than others in a high-dimensional space. The purpose of the study was to determine how visual perception depends on those six stimulus properties, and to extract detection thresholds from the latent function. Additional dataset details and an example stimulus are available in the supplementary materials.

We study how the performance of the model varies with the amount of data. For each training set size, we randomly selected a training set of that size and used the remaining data as a test set. We evaluated six models: the choice-only model, the four DDM-based variants of RT-choice with different parametric RT distributions, and the stacked RT-choice model. Model performance was measured using the expected Brier score, the expectation being over the model posterior. The Brier score [Brier, 1950] is a proper scoring rule, equivalent to the mean-squared error of predicted probability and outcome, and evaluating it in expectation over the model's posterior measures the calibration quality of the model's predictions. For each training set, performance was recorded as the difference in Brier score between each model and the choice-only baseline, to directly measure the extent to which incorporating RTs can improve the model. This evaluation was repeated for 20 random train/test folds.

Fig. 6 (left) shows results of the evaluation. All of the DDM RT-choice models performed significantly better than the choice only baseline, with the difference especially pronounced for training set sizes less than 200. The 3-parameter shifted RT distributions performed better than the 2-parameter log-normal distribution, showing the importance of having a flexible RT distribution. The stacked RT-choice model did not improve over the choice-only model. This model fails to capture important aspects of the RT distribution, such as the skew, heteroskedastic variance across the parameter space, and the presence of lapses. All of these real-world properties of RTs are captured by the DDM.

# 7 REAL-WORLD PREFERENCE LEARNING

We evaluated our model on pairwise data for preference learning using two real datasets, the first created as part of this study.

## 7.1 ROBOT GAIT OPTIMIZATION

This problem explored a 3-d space of gait parameters for a simulated quadruped robot [Rahme et al., 2020]. Using OpenAI Gym [Brockman et al., 2016], 10 second videos were recorded of gait simulations for each of 50 quasirandom points in the parameter space. A single human subject consented to data collection, and evaluated each of the 1,225 possible pairings of videos to identify which gait appeared more natural. Videos were shown side-by-side, so the subject could respond as soon as a judgement had been made. Response times were recorded for each pairing, as shown in Fig. 1. See the supplementary material for more details and example videos. The goal of the experiment was to learn the most natural-looking gait for the robot, according to the human subject.

Model evaluation was done in the same way as in the psychophysics task, by measuring the difference of expected Brier score between each model and the choice-only baseline, paired across 30 train/test folds for each training set size. Results are shown in Fig. 6 (right), and are similar to those of the psychophysics task. DDM RT-choice models significantly outperformed the choice-only model for small training sizes. Because of the smaller dimensionality of this problem ($d = 3$), the improvement fades at a smaller training set size, by around 150 observations, as the choice-only model is able to better capture the latent function in lower dimensions. The stacked model again failed to improve over choice-only.

## 7.2 RECOMMENDER SYSTEM EVALUATION

The data for this task come from a user study reported in Lin et al. [2022] in which six employees of an internet company were asked to compare pairs of A/B test results that showed performance of a recommender system under different configurations. For each pair, subjects identified which test had the better outcome, for the purpose of finding the most-preferred configuration. The results for each A/B test included changes in up to 9 metrics related to the performance of the recommender system, and the subject had to weigh the relative benefits of changes in these various metrics. The dimensionality of the configuration varied (5 to 11, median 6). We thought this experiment may help establish the limits of the benefit of the DDM, as the nature of the decision is deliberative rather than immediate: subjects reported that they had discussions with team members to help decide which option they preferred, and were evaluating the options in parallel with other tasks such as responding to messages. Consequently, the response times were substantially longer (4 seconds to over 7 minutes, median of 14 seconds—the other datasets had a *median* below

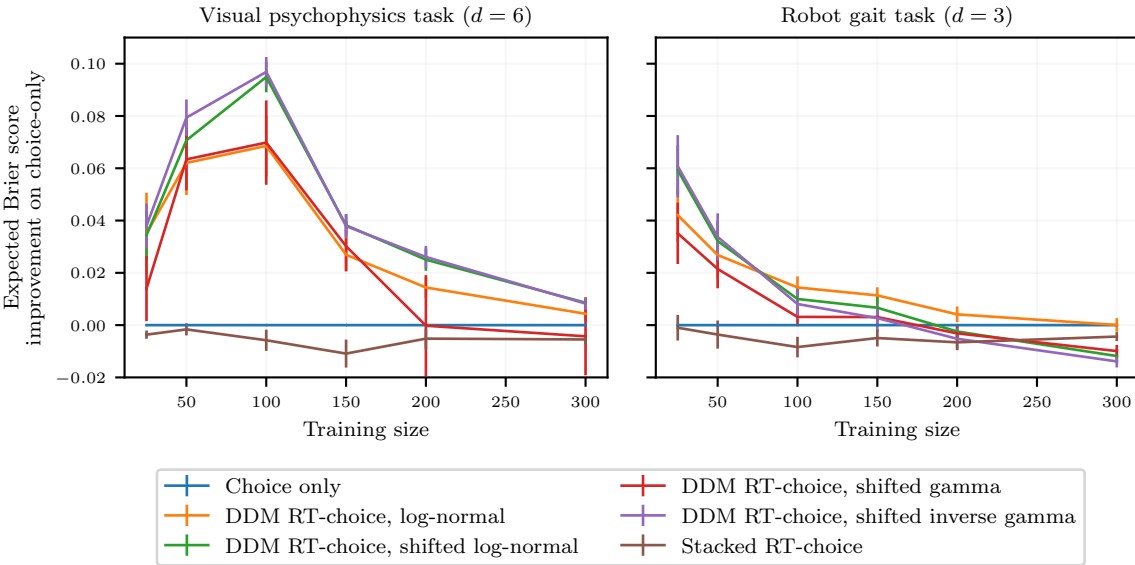

Figure 6: *Left*. Choice prediction performance on the real-world visual psychophysics dataset. Lines show the mean of the difference between expected Brier score of each model on each train/test fold, and the expected Brier score of the baseline choice-only model. Error bars show two standard errors. *Right*. Choice prediction performance measured in the same way, for the robot gait task. DDM-based RT-choice models significantly improve over the choice-only model, especially for small training sizes. The stacking approach was not able to improve over choice-only.

4 seconds). Furthermore, response times were quantized to 1s increments due to the implementation of the preference elicitation system. Additional information about the dataset is in the supplementary materials.

The amount of data in this study was insufficient to vary the training set size as in the other experiments (41 to 50 observations depending on subject). Instead, we generated 50 random splits of the data, each time training on 80% and testing on 20% to compute expected Brier score on the held-out data. Fig. 7 shows that RT-choice models outperformed the choice-only model for all six subjects. In contrast to the other two real-world experiments, our novel stacked model performed very well, significantly outperforming both choice-only and the DDM-based models in half of the subjects, and performing comparably to DDM-based models in the other half.

## 8 DISCUSSION

We have demonstrated that GP models that take into account the RT distribution improve latent function estimation and held-out predictive accuracy in both psychophysics and preference learning. By using the moments of the RT distributions provided in closed form by the DDM, we can calculate point estimates of parameters of a parametric density over RTs, and leverage this additional information to better predict human performance and understand latent cognitive representations. Our results show that measuring and modeling with RT data can improve performance across

a wide range of preference and perception learning tasks. While we focus on binary applications here, GP preference models with multinomial observation likelihoods would be a first step in extending this class of models to decision making settings with more than two options. Additional work would be necessary to identify RT distributions in a multiple-choice setting, perhaps derived from a model of the dynamics of multi-choice decision making [e.g. Roxin, 2019, Krajbich and Rangel, 2011, Tsetsos et al., 2011].

Our results also yield clear guidance for practitioners: if high-quality RTs are available (i.e. ones that are accurately measured from focused subjects), augmenting the GP choice model with the DDM improves choice prediction. If RTs do not fall into the setting where DDMs are typically productive, DDM-augmented GP models still outperform choice-only, but the more flexible stacked model may better leverage the RTs. When using the DDM-augmented model, we see that different moment-match distributions perform similarly, and the best one can be selected by cross-validation (which is feasible due to model fitting requiring only a few seconds for the training set sizes used here).

We show results in both a synthetic setting and a broad variety of real-world scenarios: human visual psychophysics, preferences in recommender system evaluation, and robot gait tuning. We note improvement specifically when the number of samples per-subject is small ($N < 200$ samples), a regime of practical utility, as it can be time-consuming and uncomfortable for humans to participate in experiments for hundreds or thousands of trials. Importantly, our model

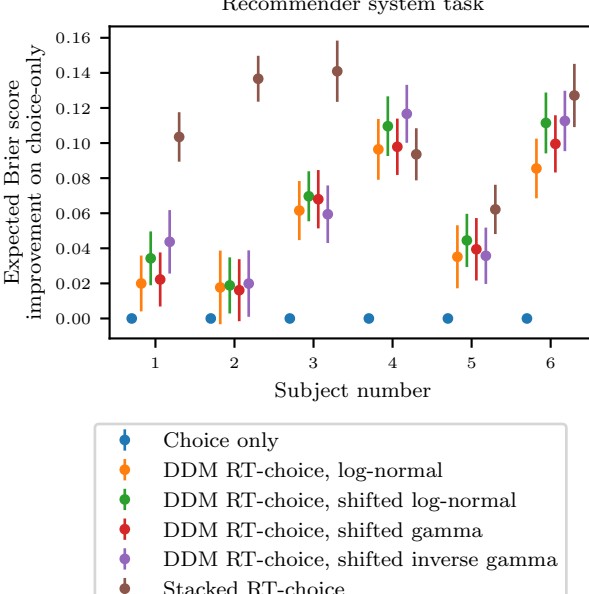

Figure 7: Expected Brier score improvement over choice-only for choice predictions in the recommender system evaluation task. For each subject, the figure shows mean and two standard errors across random train/test sets. All RT-choice models improved over choice-only, and the stacked model performed the best.

improvements are specifically salient in the small-sample regime, as the model is able to effectively leverage RT information to better estimate choice probability. As the number of samples increases, choice-only models have enough information to do very well. However, as can be seen in the left panel of Figure 6, for training sizes that are *very* small (e.g. $< 50$), the performance improvements seen in the RT model are more modest than when the number of samples is slightly increased. This suggests a 'sweet spot' for our model, where there is enough data to accurately leverage RT information, but not so much that the choice-only approaches achieve comparable performance. Incidentally, it is this regime that is likely of practical utility in many real-world human choice modelling settings.

Finally, we discuss a number of opportunities for future work. First, we focus on the benefits of our approximation for fast, differentiable inference of GP models with RTs but do not make explicit claims about the quality of approximation, which may be worse than series truncation approaches that have explicit error bounds but are challenging to apply in this setting. The interaction of approximate likelihoods (in our case, moment matching) and approximate inference (in our case, variational inference) are closest in spirit to methods for likelihood-free inference [e.g. Barthelmé and Chopin, 2014, Beaumont, 2019, Tran et al., 2017], and fu-

ture work could make this connection more explicit and provide stronger theoretical guarantees.

Second, in the stacked GP model we use a point estimate of the GP model predicting RTs, discarding the uncertainty of that model. We made this choice because integrating over this uncertainty is not possible to do in closed form, and doing so numerically would likely take our models out of the realm of practical usage with a human in the loop (an important goal for this work). Methods for propagating input uncertainty in GPs [e.g. McHutchon and Rasmussen, 2011, Villacampa-Calvo et al., 2021] provide a roadmap for the types of approximations that could be used for propagating RT uncertainty in the stacked model, but we leave that for future work.

A final opportunity for future work is pooling or utilizing of data across multiple subjects [Wiecki et al., 2013]. Combining cross-subject pooling, flexible GP models, and use of RT distributions may enable future practitioners to even better predict binary human choices in a few samples.

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

# Response Time Improves Gaussian Process Models for Perception and Preferences
# (Supplementary Material)

**Michael Shvartsman**[1]      **Benjamin Letham**[1]      **Eytan Bakshy**[1]      **Stephen Keeley**[2]

[1]Meta,
[2]Fordham University, Department of Natural Sciences, New York, NY

## A   INFERENCE DETAILS

As noted in the main text, we used standard variational methods for approximate GPs [Hensman et al., 2015, Balandat et al., 2020]. In all cases we used the Adam optimizer [Kingma et al., 2014] with a stepped learning rate beginning at 0.01 and 5000 iterations. Inputs were normalized to $[0, 1]$. The hyperprior for the lengthscale was InverseGamma$(4.6, 1.0)$, selected because it restricts approximately 95% of the prior probability mass to be between 0.1 and 0.5 (i.e. excluding very short or long lengthscales relative to the normalized input domain). The hyperprior for the variance was selected as Uniform$(1, 4)$, as it restricts the GP output to values that are not saturated by the probit sigmoid, and we wanted to keep priors consistent between the models. We additionally employed multi-start optimization (with 5 restarts), and clamped the moment-matched skew to be between 0.1 and 10 to stabilize estimation.

## B   DATASET DETAILS

### B.1   HUMAN PSYCHOPHYSICS DATASET

This dataset was obtained by contacting the authors of Letham et al. [2022] and requesting response time data for the choice dataset in the original paper. It consists of 1500 observations of a single subject making detection judgments. Stimulus features were contrast, pedestal (background luminance), temporal frequency, spatial frequency, size, and eccentricity. An example stimulus is shown in Fig. 8. Response times ranged from 0.16 to 15.87 seconds, with a median of 0.6 seconds.

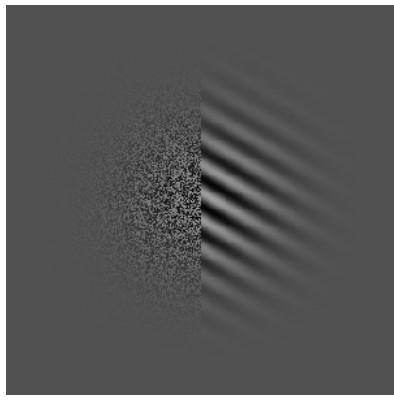

Figure 8: Example psychophysics stimulus (reprinted from Letham et al. 2022)

*Accepted for the 40th Conference on Uncertainty in Artificial Intelligence* (UAI 2024).

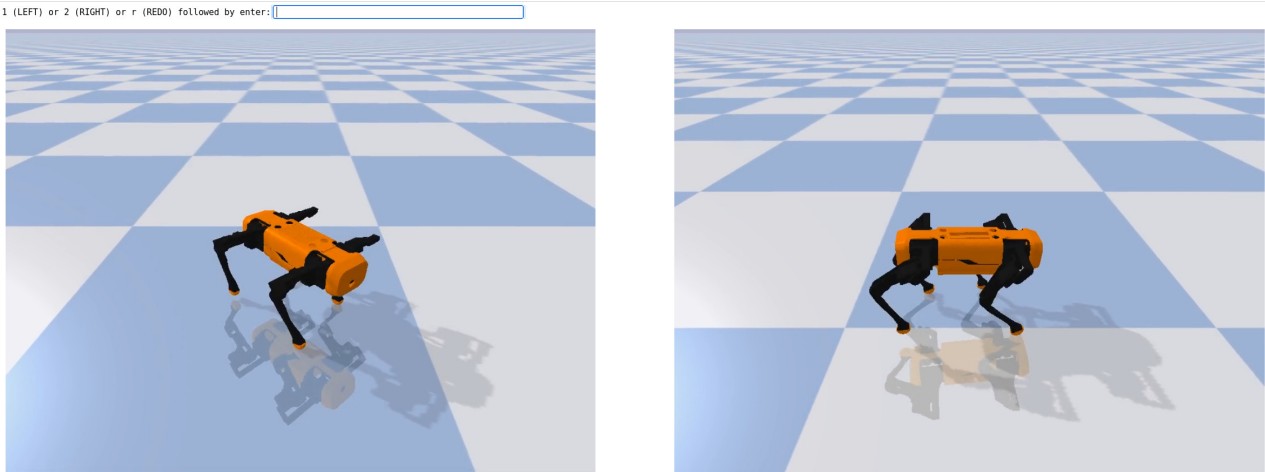

Figure 9: A screenshot of the UI for the robot gait preference learning experiment of Section 7.1. The subject viewed two videos simultaneously playing side-by-side, and selected the one with the more natural gait. Both choice and response time were recorded to fit models of gait preference.

## B.2  ROBOT GAIT PREFERENCE LEARNING

The simulation framework was from Rahme et al. [2020], and we built on the demo simulation from the package. The selected parameters and their ranges were taken from the package demo settings. Specifically, SwingPeriod ranged from 0.1 to 0.4; StepVelocity from 0.001 to 3; and ClearanceHeight from 0 to 0.1. All other gait parameters and all settings related to the simulation itself were fixed to defaults.

A total of 50 simulation videos were recorded for the study, and this supplementary material includes 3: the most-preferred, least-preferred, and median-preferred, when all videos were ranked according to the latent preference value of a 'choice-only' model fit to all of the data. The increasingly natural appearance of the gait with latent preference value is apparent. Fig. 9 shows a screenshot of the UI for the study in which the human subject viewed two gaits side-by-side and selected the more natural looking. Response times ranged from 1.54 to 9.85 seconds, with a median of 3.6s.

## B.3  RECOMMENDER SYSTEMS DATASET

This dataset was obtained by contacting the authors of Lin et al. [2022] and requesting the dataset in that paper. The dataset consisted of data from seven subjects whose response times ranged from 4 to 429 seconds. Table 1 includes additional information about this dataset. Data from subject 0 only had 20 observations, so we did not use it, since the test set size would be 4 instances only.

Table 1: Dataset details for recommender systems dataset.

| subject ID | Instances | Dimensions | Minimum RT (s) | Maximum RT (s) | Median RT (s) |
|---|---|---|---|---|---|
| 0 | 20 | 7 | 5 | 46 | 11 |
| 1 | 41 | 8 | 11 | 429 | 30 |
| 2 | 41 | 11 | 6 | 250 | 14 |
| 3 | 43 | 5 | 7 | 159 | 14 |
| 4 | 50 | 6 | 7 | 118 | 15 |
| 5 | 50 | 7 | 4 | 157 | 7 |
| 6 | 50 | 8 | 4 | 35 | 9 |

# C  MOMENT MATCHING RESULTS

Here we provide the probability density functions of all heavy-tailed reaction time distributions we use in this work. Of these, the log-normal is the only two-parameter distribution whereas the shifted gamma, shifted inverse gamma and the shifted log-normal are all three-parameter distributions. The sample statistics calculated from the reaction times, specifically the mean, variance, and skew, are denoted $m_*$, $v_*$, and $s_*$, respectively. Expressions for the three-parameter distributions below are adapted from Lo et al. [2014].

## C.1  SHIFTED LOG-NORMAL

$$f(z; \mu, \sigma, \eta) = \frac{1}{\sigma(z - \eta)\sqrt{2\pi}} \exp\left\{-\frac{(\ln(z - \eta) - \mu)^2}{2\sigma^2}\right\}, \quad z > \eta$$

with parameter estimates as a function of sample statistics

$$\hat{\mu} = \ln(m_* - \eta) - \frac{\sigma^2}{2}, \quad \hat{\sigma^2} = \ln\left|1 + \frac{v_*}{(m_* - \eta)^2}\right|, \quad \hat{\eta} = m_* - \frac{\sqrt{v_*}}{s_*}\left[1 + (B)^{\frac{1}{3}} + (B)^{-\frac{1}{3}}\right]$$

$$B \equiv \frac{1}{2}\left(s_*^2 + 2 - \sqrt{s_*^4 + 4s_*^2}\right) \in (0, 1].$$

## C.2  SHIFTED INVERSE GAMMA

$$f(z; \alpha, \beta, \eta) = \frac{\beta^\alpha}{\Gamma(\alpha)}\left(\frac{1}{z - \eta}\right)^{\alpha - 1} \exp\left\{-\frac{\beta}{z - \eta}\right\}, \quad z > \eta, \beta > 0$$

with parameter estimates as a function of sample statistics

$$\hat{\eta} = m_* - \frac{\sqrt{v_*}}{s_*}\left[2 + \sqrt{4 + s_*^2}\right]$$

$$\hat{\alpha} = 2 + \frac{(m_* - \eta)^2}{v_*}$$

$$\hat{\beta} = (m_* - \eta)(\alpha - 1).$$

## C.3  SHIFTED GAMMA

$$f(z; \alpha, \beta, \eta) = \frac{(z - \eta)^{\alpha - 1}}{\beta^\alpha \Gamma(\alpha)} \exp\left\{-\frac{z - \eta}{\beta}\right\}, \quad z > \eta, \beta > 0$$

with parameter estimates as a function of sample statistics

$$\hat{\alpha} = \frac{4}{s_*^2}, \quad \hat{\beta} = \sqrt{\frac{v_*}{\alpha}}, \quad \hat{\eta} = m_* - \alpha\beta.$$

# D  THE DDM LINK FUNCTION

Figure 3 showed that, depending on the parameters, the choice probability link function implied by the DDM can closely match either the logistic or probit links. Fig. 10 shows a similar comparison using the actual DDM parameters from models fit to the data in the experiments. For the visual psychophysics and robot gait tasks, we fit a model with a randomly sampled training set of size 300, and used the shifted log normal matched DDM distribution. The fitted link functions are given in the figure, alongside the logistic and probit link functions.

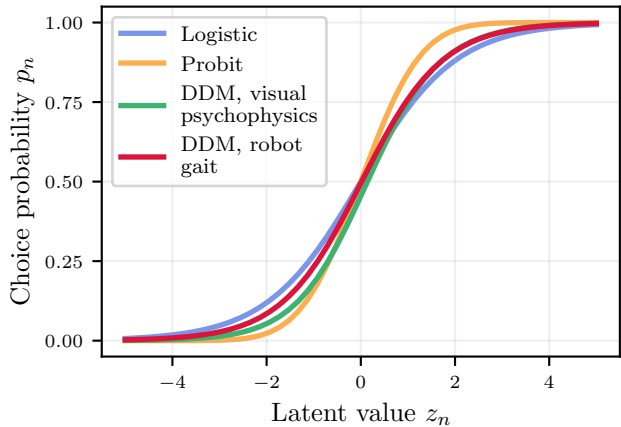

Figure 10: The DDM link function, with parameters fit to the data from the visual psychophysics and robot gait tasks. The link functions are similar though not identical to the logistic and probit link functions.

# E  ETHICS STATEMENT AND BROADER IMPACTS

Our work carries low-risk of ethical harm, as it focuses on binary responses in simple decision-making tasks in low-sensitivity settings. For this work, we only consider de-identified data where subjects provided explicit informed consent, and we keep our conclusions focused on model performance. We draw no broad conclusions about general human behavior. We anticipate minimal risk associated with future application of our work.

# F  COMPUTATIONAL LOAD

The methods developed in this paper are not computationally demanding. All benchmarks were run on a standard laptop computer. Single model fits, which are most relevant for future practitioners, take on the order of seconds on a typical laptop.

