# OpenReview forum: "Response Time Improves Gaussian Process Models for Perception and Preferences"
_auai.org/UAI/2024/Conference — UAI 2024 poster_

### Official Review · Reviewer_zL1x · 2024-03-18

**Q2-1 Originality-Novelty:** 2
**Q2-2 Correctness-Technical Quality:** 3
**Q2-5 Clarity Of Writing:** 4

**Q10 Ethical Concerns:**

yes. Small paragraph in appendix indicates the low-risk nature of this work.

**Q1 Summary And Contributions:**

This paper incorporates response times taken during decision making processes into a diffusion decision model to better model human preferences. It is assumed that response times are a proxy for difficulty in making a preference.

**Q2-3 Extent To Which Claims Are Supported By Evidence:**

3: Good: the main claims are supported by convincing evidence (in the form of adequate experimental evaluation, proofs, (pseudo-)code, references, assumptions).

**Q2-4 Reproducibility:**

2: Fair: key resources (e.g. proofs, code, data) are unavailable but key details (e.g. proof sketches, experimental setup) are sufficiently well-described for an expert to confidently reproduce the main results.

**Q3 Main Strengths:**

The primary strengths of this work are the ability efficacy of results and author's description of when to use the described methods. They also develop an implementation which works well on standard laptops.
The clarity of writing is also a major strength in this work.

**Q4 Main Weakness:**

I am not familiar with decision models and I can not point to any major issues in this articulate paper.

**Q5 Detailed Comments To The Authors:**

My only criticism would be to investigate the very small uncertainty estimates when using the stacked RT-choice. It is only just possible to see uncertainty in the Expected Brier score improvement on choice-only (figure 7) for subject 3. I would be interested to understand why this method is yielding less uncertainty than the other proposed methods.

**Q9 Complying With Reviewing Instructions:**

Yes

---

> ### Author Rebuttal · Authors · 2024-04-06
>
> We thank the reviewer for the very positive feedback of our contribution! Regarding the error bars on Fig. 7, we have a few clarifications: the error bars are defined over paired differences relative to the choice-only model (hence the choice-only model has no error bar, since its paired difference to itself is always 0). Regarding the other models, we did find that the DDM-RT models were more difficult and somewhat more variable in fitting (they required some stabilizations heuristics such as clipping moments and multi-start optimization), hence those error bars are larger than those for the stacked GP model. We ran the cross-validation on a sufficiently large number of folds (100) to reliably show the differences among models if they do exist, so it is not surprising that the model with the more stable fits will show smaller error bars.

---

### Official Review · Reviewer_7E4o · 2024-03-20

**Q2-1 Originality-Novelty:** 3
**Q2-2 Correctness-Technical Quality:** 3
**Q2-5 Clarity Of Writing:** 2

**Q10 Ethical Concerns:**

no concerns

**Q1 Summary And Contributions:**

This paper proposes two approaches for incorporating response time (RT) into Gaussian process (GP) preference and perception models. The first is based on stacking GP models, and the second uses a novel differentiable approximation to the likelihood of the diffusion decision model (DDM), the de-facto standard model for choice RTs.

These RT-choice GPs are testified to enable better latent value estimation and held-out choice prediction relative to baselines.

**Q2-3 Extent To Which Claims Are Supported By Evidence:**

2: Fair: the main claims are somewhat supported by evidence (but the experimental evaluation may be weak, or does not match entirely with the claims, important baselines may be missing, proofs contain important ideas but lack rigor, algorithmic details are only discussed superficially, references are imprecise, assumptions are not sufficiently motivated or explicated, etc.).

**Q2-4 Reproducibility:**

2: Fair: key resources (e.g. proofs, code, data) are unavailable but key details (e.g. proof sketches, experimental setup) are sufficiently well-described for an expert to confidently reproduce the main results.

**Q3 Main Strengths:**

1. as the key contribution, the approximation of the DDM (drift-diffusion model) using a family of parametric skewed distributions, enables the usage of GP models with DDM-inspired RT-choice likelihoods.

**Q4 Main Weakness:**

1. as mentioned in this paper, single subject at a time is considered as a shortage and multiple subjects in parallel at a time is frequent in the real-world.
2. the comparisons with baselines (such as Figure 7) contain less models from outside and this makes it relatively difficult to judge the overall importance of this paper's methods at a high-level.

**Q5 Detailed Comments To The Authors:**

1. can you append more baselines (such as append to Figure 6,7) for evaluating of your proposals.

**Q9 Complying With Reviewing Instructions:**

Yes

---

> ### Author Rebuttal · Authors · 2024-04-06
>
> We thank the reviewer for their positive assessment and helpful comments. We address their specific questions next:
> * Regarding the limitation to a single participant: we agree that extensions to multiple participants are interesting, but we think the ultimate goal of work such as ours is always personalized models that can run with the human in the loop. In the case of psychophysics, data is typically collected from one participant at a time even if it is aggregated for analysis later. In the case of preference learning, aggregation of preferences across individuals is not typically considered (or if it is, it leads to novel contributions and publications in their own right, e.g., Houlsby et al. NeurIPS 2012, Khan et al AISTATS 2014).
> * Regarding additional baselines: we would love to find additional baseline approaches to compare our methods to, but it appears that general modeling of perception and preferences defined over arbitrary multidimensional and continuous stimulus spaces is relatively nascent. In fact, the stacked GP approach we propose was added largely to introduce additional baselines to the field. We would happily evaluate our model against any additional approaches the reviewer can point us to.
> * Regarding train / inference time cost: we mention in the supplement that a single run of estimation takes on the order of seconds on a typical laptop, and will surface this information earlier in the text.

---

### Official Review · Reviewer_YABj · 2024-03-21

**Q2-1 Originality-Novelty:** 2
**Q2-2 Correctness-Technical Quality:** 3
**Q2-5 Clarity Of Writing:** 3

**Q1 Summary And Contributions:**

The authors propose two approaches for incorporating response times (RTs) into preference GP models. The first considers the diffusion decision model (DDM), a standard model for choice RTs, which is incorporated into preference model through joint modelling with differentiable likelihood for DDM. The second approach fits independent GP on response times data, and uses its predictive mean as a feature in preference GP. They demonstrated improved performance over vanilla GP preference model in small data regime in synthetic and real-world datasets.

**Q2-3 Extent To Which Claims Are Supported By Evidence:**

3: Good: the main claims are supported by convincing evidence (in the form of adequate experimental evaluation, proofs, (pseudo-)code, references, assumptions).

**Q2-4 Reproducibility:**

3: Good: key resources (e.g. proofs, code, data) are available and key details (e.g. proofs, experimental setup) are sufficiently well-described for competent researchers to confidently reproduce the main results.

**Q3 Main Strengths:**

Since response times do not require active querying of the user, the idea is well-motivated as long as performance improvement over the vanilla preference learning approach can be demonstrated (which I think is convincing enough in Sections 5-6).

**Q4 Main Weakness:**

Novelty for probabilistic ML is somewhat limited. Particularly, the stacking approach appears straightforward. Why is the uncertainty of the response time model being ignored? However, inference with the DDM model is not as straightforward, and it seems that the adopted moment matching approach performs well. Nonetheless, it is not clear to me what the implications of the chosen inference approach are, as it combines moment matching and variational inference. It appears that the response time likelihood is 'approximated' through moment matching, which is then used in the ELBO: "This moment-matched parametric distribution is then used as the RT component of the likelihood in (2), p(tn |zn , yn , θD )"
If this evaluation is correct, what guarantees exist for the inference?

**Q5 Detailed Comments To The Authors:**

"...we assume that the choice probability is determined by the difference in utility between the two choices [Chu and Ghahramani, 2005]. Since GPs are closed under addition, the prior over utility differences remains a GP with a ‘preference kernel’ given by Houlsby et al. [2011]. In both cases..." Do you cast preference learning as classification by modelling the utility difference as a GP, or do you adopt the approach taken by Chu et al. where GP is fitted to the utility function?

"Model performance was measured using the expected Brier score, the expectation being over the model posterior. The Brier score [Brier, 1950] is a proper scoring rule, and evalu- ating it in expectation over the model’s posterior measures the calibration quality of the model’s predictions." For reader's convenience, it would be nice to recall what this Brier score is (MSE between prob predictions and outcomes).

Can you analyse why stacking approach is so performing so well in Experiment 7.2?

**Q9 Complying With Reviewing Instructions:**

Yes

---

> ### Author Rebuttal · Authors · 2024-04-06
>
> We thank the reviewer for their detailed comments and overall positive feedback. We address the individual points raised next:
> * Regarding the uncertainty of the RT model in the stacked GP setting: we agree with the reviewer that propagating other features of the RT distribution like uncertainty in the stacked-GP model could improve performance. However, integrating over this uncertainty is not possible to do in closed form, and doing so numerically would likely take our models out of the realm of practical usage with a human in the loop (an important goal for this work). Methods for propagating input uncertainty in GPs (for example, Villacampa-Calvo et al., “Multi-class Gaussian process classification with noisy inputs”, JMLR 2021) provide a roadmap for the types of approximations that could be used for propagating RT uncertainty in the stacked model, but we leave that for future work..
> * Regarding the interaction between the likelihood approximation and the inference algorithm: the reviewer is correct that the interaction between the two approximations is not theoretically well-understood. The variational approximation has the usual guarantees w.r.t. the family of skewed approximating distributions, but not w.r.t. the intractable full DDM likelihood, and to our knowledge there is no general understanding of the relationship between matching moments and likelihood-based inference. We highlight that the combination of approximate likelihood and approximate inference is not unusual in cases when exact likelihoods are intractable (e.g. Barthelmé & Chopin 2011; Gutmann & Coriander 2016), though unlike those settings our approach is fast and does not rely on an expensive surrogate or simulator (at the cost of being specialized to one family of likelihoods).
> * Regarding the preference learning problem: we fit a GP to the utility as in Chu & Gharhamani, which implies that the utility difference is also a GP with the kernel given by Houlsby et al.
> * Regarding the Brier score, we take the reviewer’s point that reminding the reader of its definition and properties would be useful, and will address this in the camera-ready version.
> * Regarding the benefit of the stacking approach, we hypothesize that the relative benefit of stacking vs. the DDM likelihood will depend on which model is better specified. The stacking model assumes, among other things, Gaussian observation noise for the log RT. The DDM model assumes a decision process that, as discussed in the paper, is best suited for more rapid and non-deliberative decisions and as such likely inconsistent with the long response times in the recommender system problem. In practice, model selection via cross validation will be a reliable approach for determining the appropriateness of the model in any given setting.

---

### Official Review · Reviewer_CCgC · 2024-03-22

**Q2-1 Originality-Novelty:** 3
**Q2-2 Correctness-Technical Quality:** 2
**Q2-5 Clarity Of Writing:** 3

**Q1 Summary And Contributions:**

The paper presents two novel approaches to incorporate response time (RT) into preference learning with a Gaussian process (GP) model. The authors propose a method that combines a choice likelihood with a diffusion decision model (DDM) likelihood. Since the DDM likelihood is not usually differentiable, the authors use a moment approximation of the RT density. The moments of the RT density are available in analytical form with differentiable expressions. The authors also propose a simpler stacked approach where inference is done with two GP models one for the response time and another one for the choices. In this approach, the sub-models are then stacked and inference is done jointly. The two main models are benchmarked against state-of-the-art on synthetic and real-world data. The results show that incorporating RT improves  performances. The experiments also show that a stacked model might perform better when response times are less reliable.

**Q2-3 Extent To Which Claims Are Supported By Evidence:**

3: Good: the main claims are supported by convincing evidence (in the form of adequate experimental evaluation, proofs, (pseudo-)code, references, assumptions).

**Q2-4 Reproducibility:**

3: Good: key resources (e.g. proofs, code, data) are available and key details (e.g. proofs, experimental setup) are sufficiently well-described for competent researchers to confidently reproduce the main results.

**Q3 Main Strengths:**

- The idea of incorporating RT is important and easily justified from a theoretical point of view
- the moment based approximation of the likelihood seems good enough
- the experimental section shows that RT plays an important role in improving the performances

**Q4 Main Weakness:**

- The use of the moment derived in (Srivastava et al. 2016) is relatively clear, however I would have appreciated a bit more details.
- More in general there are several points in section 4 that could be better explained. See comments below
- In the experimental section, there are some results that require more explanation.

**Q5 Detailed Comments To The Authors:**

- Regarding section 4:
    - The moment formulae are taken directly from (Srivastava et al. 2016), however a bit more explanatins could make this paper a bit more self-contained. The functions coth and csch are rather standard, however I think that either a definition in appendix or at least the name written in long form could help readability. Moreover, it would greatly help to have a sentence linking the variables used in Srivastava et al. 2016 (k_z, k_x) with the variables used in this paper (k_z and y_n).
    - On the link function defined in eq. (3), is this the only possible choice? Have the authors also thought about other link functions? If not, why not?
    - Variational approximations are often affected by convergence issues. Have the authors experienced such issues? Especially for the parameters of the likelihood in the RT density approximation this could be an issue, could the authors comment on this aspect?
- Regarding the experimental section:
    - the synthetic experiments are evaluated on MSE scores, however, I might be mistaken here, I think that the latent function estimated by the GP should be scale independent (what matters are the relative differences), so I am wondering if these scores are still meaningful.
    - The authors already give a convincing explanation on why the staked GP performs much better on the last experiment than on the other ones. However I am wondering if this effect is just a matter of reliability of the RT or also a matter of scale. In the examples where the stacked GP does not improve over choices, the RT are very small, could this mean that RT does not impact performances simply because \tilde{t}_n in the second model has a small impact? Are the authors using an ARD kernel for this second model?
    - In fig. 6 left, why does the improvement seem to increase initially for the DDM RT models?

**Q9 Complying With Reviewing Instructions:**

Yes

---

> ### Author Rebuttal · Authors · 2024-04-06
>
> We thank the reviewer for their helpful comments and positive outlook on our work. We address the detailed questions and comments directly:
> * We agree that the use of different variable names is somewhat confusing. We had intended to make the linkage between the DDM drift parameter and the latent GP explicit by using y_n but did not emphasize the connection back to the Srivastava et al. expressions. We will address this in the revised manuscript and include a table in the appendix specifying the variable differences between us and Srivastava et al. We will additionally be sure to explicitly mention the hyperbolic cosecant and cotangent functions in the main text, as we agree they are somewhat uncommon.
> * Regarding the link function: yes, expression 3 is the only possible choice for the DDM-RT model due to the underlying model structure (i.e. it is the expression for choice probability given the DDM dynamics, without approximation). Other link functions might be possible by changing that underlying model (e.g. if it were an Ornstein-Uhlenbeck process instead of a Wieiner process), but the standard link functions (e.g. logit/probit) are not derived from an underlying DDM, so are not appropriate here. We could consider a different link for our stacked-RT model, as here the link not derived from the DDM. However, we suspect this will have very minimal effects given the close relationship between the DDM link and the canonical links for particular parameter settings, as shown in figure 3.
> * Regarding convergence issues: we appreciate the authors’ insight on the practicalities of working in this setting. We have experienced some fit instabilities largely addressed by a number of features of our codebase (which we plan to release):
>     * The use of stochastic optimizers which are more robust to numerical issues, with multiple restarts to handle fit failures and pathologies.
>     * Clipping moments, especially skew, to values for which likelihoods are defined and relatively well-behaved.
> We will explicitly discuss these points in a revised version of the manuscript.
> * Regarding the use of Brier score, we emphasize that it is MSE on the predicted *choice probability*, not latent function, and furthermore that even the latent function estimated by the GP is only scale-independent in the paired preference setting but not in the single-choice setting. In our setting, the scores are not only well-defined but in fact measure the quality of our models’ calibration (Brier, 1950).
> * Regarding the scale of the RTs in the stacked GP model: we always both standardize the inputs to the GPs and use an ARD kernel, such that scale differences on RT are unlikely to be causing these differences.
> * Regarding fig 6: Early on, insufficient data is available to characterize RTs sufficiently well to give our models their full benefit, and later on enough data is available for the choice-only reference model to catch up, thus creating this peak. A similar peak is present for even small data sizes in the lower-dimensional gait preference data; the figure in the submission uses a smallest training size of 25, but if that is dropped to 5 then that problem also shows the same behavior (https://imgur.com/a/7qoIGhl)

---

### Official Review · Reviewer_aF8f · 2024-03-26

**Q2-1 Originality-Novelty:** 2
**Q2-2 Correctness-Technical Quality:** 3
**Q2-5 Clarity Of Writing:** 3

**Q1 Summary And Contributions:**

This paper investigates modelling the response time together with choice through Gaussian processes. In particular, they use the wiener process to model such situations, which is very natural. Fig. 2B gives clear illustration about the approach. It is great to see the authors are able to use previous results to find the momental estimation for the response time. They authors have used synthetic data and real data results to evaluate their models.

**Q2-3 Extent To Which Claims Are Supported By Evidence:**

3: Good: the main claims are supported by convincing evidence (in the form of adequate experimental evaluation, proofs, (pseudo-)code, references, assumptions).

**Q2-4 Reproducibility:**

3: Good: key resources (e.g. proofs, code, data) are available and key details (e.g. proofs, experimental setup) are sufficiently well-described for competent researchers to confidently reproduce the main results.

**Q3 Main Strengths:**

The main strengths of the paper is:

**1**, the use the wiener process to model the choice and the response time is very interesting and natural.

**2**, the momental estimation of the response time is good.

**Q4 Main Weakness:**

The weakness of the paper is:

**1**, the experimental evaluation seems a bit weak as there seems to be limited comparison methods' results.

**Q5 Detailed Comments To The Authors:**

In general, I like the construction of the idea of using wiener process to model the choice and the response time.

one question, what if we have more than two choices in the preference model?

**Q9 Complying With Reviewing Instructions:**

Yes

---

> ### Author Rebuttal · Authors · 2024-04-05
>
> We thank the reviewer for their helpful comments and overall positive assessment of our work.  The reviewer commented on the limited number of methods considered in our work.  Unfortunately, there are few models for general multi-dimensional psychophysics and preference learning (rather than specific parametric forms for settings such as audiometry), and even fewer that consider the addition of response times. This limits the number of baselines we can compare against, which is what motivated us to  introduce the stacked GP model. This introduces an  additional competitive baseline against which our contribution can be compared.
>
> Regarding preference models with more than two choices: the model as it currently stands already covers preferences over arbitrary sets of items with arbitrary numbers of dimension / attributes, as long as comparisons are binary. If instead the decision maker decides between more than two options, additional information is needed to define the task (e.g. ranking multiple options, selecting one vs the rest, etc) and how this task is modeled. A variety of these problems are studied in the choice-only setting. We briefly speculate on how such extensions would look for the RT model:
> * For the DDM RT model, the model as we use it is only theoretically motivated for binary choices, but could still be used empirically in any setting where we have a latent scalar value that is converted to both a binary choice and a response time (e.g. in selecting one alternative vs a set of others).
> * On the stacked-GP front, likewise the model should function in any case where scalar RTs can be predicted from stimulus configurations, and used in the choice GP.

---

### Meta-Review · Area_Chair_8kEW · 2024-04-17

This paper tackles an interesting problem, that incorporating user response time into preference learning, and brings a number of interesting and relevant techniques to the table to improve GP models. A nice conceptual contribution, with non-trivial novelty and some decent analysis and empirical evaluation. All of the reviewers were positive on the paper to varying degrees. That said, the reviewers posed some questions and made several suggestions that should be used to improve the paper.

All in all, this should make a very nice contribution to UAI.